# Cultivating capacities in community-based researchers in low-resource settings: Lessons from a participatory study on violence and mental health in Sri Lanka

**Alexis Palfreyman**[1]*, **Safiya Riyaz**[2], **Zahrah Rizwan**[2], **Kavitha Vijayaraj**[2], **I. P. R. Chathuranga**[2], **Ruwanka Daluwatte**[2], **W. A. T. Devindi**[2], **B. Shakila Eranda**[3], **Vinodani Jayalath**[3], **Azam Junaid**[2], **Ashra Kamal**[2], **Shehan Koshila Kannangara**[3], **K. M. G. Prasanga Madushani**[3], **Laksi Mathanakumar**[4], **S. Ihjas Mubarak**[2], **Vithusha Nagalingam**[5], **Sachin Palihawadana**[2], **Ravishanka Pathirana**[2], **V. G. Sameera Sampath**[3], **Lojini Shanmuganathan**[2], **Tharindi Thrimawithana**[3], **Priyatharshiny Vijayaratnam**[4], **Sasith Lakshan Vithanage**[3], **R. K. K. A. Sajini Wathsala**[3], **R. Mervin Yalini**[4]

1 Institute for Global Health, University College London, London, United Kingdom, 2 Independent Researcher, Colombo, Sri Lanka, 3 Independent Researcher, Galle, Sri Lanka, 4 Independent Researcher, Jaffna, Sri Lanka, 5 Independent Researcher, Batticaloa, Sri Lanka

* a.palfreyman@ucl.ac.uk

**Data Availability Statement:** Our dataset is not open access to ensure anonymity of parent study

## Abstract

Participatory methods, which rely heavily on community-based data collectors, are growing in popularity to deliver much-needed evidence on violence and mental health in low- and middle-income countries. These settings, along with local researchers, encounter the highest burden of violence and mental ill-health, with the fewest resources to respond. Despite increased focus on wellbeing for research participants and, to a lesser degree, professional researchers in such studies, the role-specific needs of community-based researchers receive scant attention. This co-produced paper draws insights from one group's experience to identify rewards, challenges, and recommendations for supporting wellbeing and development of community-based researchers in sensitive participatory projects in low-resource settings. Twenty-one community-based researchers supporting a mixed-methods study on youth, violence and mental health in Sri Lanka submitted 63 reflexive structured journal entries across three rounds of data collection. We applied Attride-Stirling's method for thematic analysis to explore peer researchers' learning about research, violence and mental health; personal-professional boundaries; challenges in sensitive research; and experiences of support from the core team. Sri Lanka's first study capturing experiences of diverse community-based researchers aims to inform the growing number of global health and development actors relying on such talent to deliver sensitive and emotionally difficult work in resource-limited and potentially volatile settings. Viewing participatory research as an opportunity for mutual learning among both community-based and professional researchers, we identify practice gaps and opportunities to foster respectful team dynamics and create generative and safe co-production projects for all parties. Intentional choices around communication, training, human and consumable resources, project design, and navigating

participants who are (closely) socially connected to the study's authors and may be re-identified through this study's underpinning data. This is in keeping with standards for consent set forth by the ethics approval granted for this study. Requests for further information about data availability can be made to ethics@ucl.ac.uk.

**Funding:** This research was funded by the UK National Institute for Health and Care Research (NIHR) (GHR 17/63/47) using UK aid from the UK Government to support global health research. This NIHR Award funded the participation of all authors in this study. The funders had no role in study design, data collection and analysis, decision to publish, nor preparation of this manuscript. The views expressed in this publication are those of the authors and not necessarily those of the NIHR or the UK Department of Health and Social Care.

**Competing interests:** The authors have declared that no competing interests exist.

instable research conditions can strengthen numerous personal and professional capacities across teams. Such individual and collective growth holds potential to benefit short- and long-term quality of evidence and inform action on critical issues, including violence and mental health, facing high-burden, low-resource contexts.

## 1. Introduction

High quality research on violence and mental health remains deficient but essential to advance evidence and action for low- and middle-income countries (LMIC) harboring the greatest burden, yet fewest resources to respond [1]. Participatory methods, which rely heavily on community-based researchers (CBRs), are growing in popularity to deliver such critical research, bringing rewards and challenges for research teams and affected communities [2, 3]. While research institutions, funders, and ethics bodies increasingly encourage practice standards to support wellbeing among participants in (participatory) violence and mental health studies [4, 5], comparable efforts to understand and advocate for attending to the role-specific needs of researchers, and *especially* CBRs, are lacking [6, 7]. Given the legacy of imbalanced power relations "in the creation of global health, and how [it] persists in the field" [8, p1059], concerted efforts to value the expertise and capacities of local collaborators like CBRs and communities are paramount in contemporary global health research [2]. This co-produced paper responds to current deficiencies in global health scholarship to make recommendations for supporting wellbeing and development of CBRs in sensitive participatory research in low-resource settings.

CBRs are members of the target population and often new to research before training as co-researchers [9]. Their insider status has shown advantages for investigating sensitive topics and potentially vulnerable populations, including those affected by violence and mental ill-health [3, 10, 11]. The more balanced power dynamics between CBRs and community participants may facilitate inclusion of groups less accessible to external researchers [2]. CBRs also bring situated knowledge to the research process, potentially improving its transparency [12] and quality through enabling richer insights and relevance of research outputs [13] and identifying and developing more context-appropriate interventions [14]. Limited evidence also purports personal and social benefits to CBRs including capacity strengthening, increased knowledge and earnings, and personal and community empowerment [12–14].

Alongside any potential benefits for research quality and CBRs themselves, however, their unique responsibilities in participatory projects may also present personal and professional risks. CBRs play multiple and competing roles, complicating their position during the research cycle [15]. Intrinsic challenges of qualitative research around confidentiality, anonymity and researcher-participant relationships are compounded in participatory projects when researchers collect data from people they know [16]. Evidence from LMIC researcher experiences, including CBRs, specifically cites concerns around personal safety, logistics, emotional distress, role conflict and pre-existing participant relationships, all of which can impact perceived objectivity and wellbeing [15]. While few studies explore power dynamics in participatory projects between CBRs and primary or otherwise professional investigators, CBRs' are likely less able to voice dissenting views and challenge decision-making on behalf of themselves and their communities, especially in lower-resource settings [2]. Power also converges with matters of representation within participating communities when considering who can take up CBR roles, with risks of reproducing exclusion of those most affected by a given issue in the absence

of conscious recruitment [17]. Beyond matters of power, the sensitivity of studying violence and mental health itself may introduce additional strains on CBRs' wellbeing and that of their wider teams, as observed in other professions encountering trauma [7]. Yet despite concerns that, "participatory research [may expose] different safeguarding risks compared with traditional research approaches" [2, p12] few studies have explored the experiences of CBRs [14, 15, 18, 19] or the impact participatory research models have on them [3].

Considering calls to better account for power, training, and resource differentials among global health and violence research partners [15, 19, 20], we captured experiences of CBRs from one participatory project on violence and mental health in Sri Lanka, guided by four research questions:

1. What learning occurred across the research cycle amongst CBRs?

2. How did CBRs' pre-existing relationship with study participants impact their role?

3. What challenges did CBRs face, if any?

4. How did CBRs navigate challenges and what support, if any, did they want from the core research team?

We identify concrete practice gaps and possibilities, by priority area and research phase, to establish safe and mutually generative team dynamics that may foster thrivability in CBRs and reduce risks of harm to LMIC co-researchers and communities in sensitive participatory research [2]. This study provides an advocacy resource for future researchers seeking improved conditions in similar projects.

## 2. Materials and methods

### a. Study setting and scholarly context

This study and all authors were nested within a larger multi-method Participatory Ethnographic Evaluation and Research (PEER) project exploring violence, parent-youth dynamics, COVID-19 and their relationship with youth mental health in Sri Lanka (NIHR award 17/63/ 47). Evidence on exposure to different forms of violence against children and youth in Sri Lanka remain woefully deficient [21]. Despite Sri Lanka's often regionally impressive health and education indicators [22], youth mental health, including in the context of violence and now COVID-19, are a growing concern [23]. Sri Lanka continues to observe annual increases in non-fatal self-harm, disproportionately affecting adolescent girls [24], while the legacy of civil conflict (1983–2009) differentially impacts youth along ethnic, geographic and socioeconomic lines.

The authors of this study were not involved in securing the grant through which this and the parent PEER study were made possible. COVID-19, having disrupted original grant plans, introduced an opportunity to explore pandemic-safe alternatives with approval from the funder, grant colleagues, and ethics boards. With a retained but broad goal to explore violence and mental health among youth, the in-country primary investigators–hereafter referred to as the core team–wished to include the views of a diverse cross-section of youth in the role of CBRs. Conscious recruitment through popular local employment platforms; professional, activist, youth (parliament), community-based and educational networks; and social media identified CBRs. Core team members independently and jointly explored candidate participatory methods and presented them in a 'methods showcase' to discuss options. PEER was identified as a base approach that could permit COVID-safe participation of all team members and be modified to accommodate CBR input. In keeping with the PEER approach [25], our CBRs were recruited from the same social networks as participants and actively participated in all

stages of the parent research cycle from topic selection through dissemination. Through interactive exercises, CBRs chose (i) study foci, (ii) how to work together, (iii) study participants, and (iv) gave feedback on timelines, protocols, and practicalities. The core team collaborated alongside CBRs to shape (v) whether and how to feasibly incorporate additional creative methods, (vi) design of study materials, and (vii) dissemination strategy and outputs, of which this is one. Between March and October 2021, CBRs conducted three iterative rounds of data collection using in-depth interviews, autophotography and creative writing through Story Completion Method. Rounds were punctuated by individualised inter-round debriefs and preliminary team-based data analysis. The current study was conducted concurrently with the PEER project, amid COVID-19. As such, both studies followed Sri Lankan government protocols and personal and participant safety preferences, adjusting activity modes with pandemic conditions.

## b. Study sample, team positionalities and reflexivity

CBR eligibility criteria included being 18–29 years old at the start of the study; residing in Eastern, Northern, Southern or Western Provinces to support varied perspectives; being able to commit to active involvement in the full project cycle; and successful completion of the foundational PEER training programme. Twenty-one CBRs were recruited, and all completed their duties. While CBRs contributed anonymised feedback for internal project learning, all 21 also chose to generate these data as external collaborative knowledge (i.e., this study) alongside primary investigators. Table 1 presents CBRs' background characteristics against an overall national profile where possible (*) [26]. Contemporary national data disaggregated to explore youth (18–29) characteristics including marriage, family and socioeconomic factors are unavailable. Our CBR sample over-represents minority youth proportional to national demographics, particularly Tamils, due to the parent project's intentional inclusion of often invisibilised communities from previous research. CBRs are also atypical in their near-universal tertiary education as just 3% of the population holds a degree or higher [26], but still represent mixed community backgrounds seldom supported to lead local research efforts. More women (57%) participated than men (43%), above national proportions (52% women, 48% men) [26], but mirroring trends in violence research globally, possibly due to gendered norms, topic perceptions, and greater proximity to survivorhood [7].

The core team has supported multiple community-based research projects through Sri Lanka's non-profit sector inclusive of youth, violence, and mental health studies (ZR, KV, and SR). AP has led multi-method research on violence and health in Sri Lanka for more than a decade. Together, all research experience and educational levels, languages, ethnicities, religions, genders, socioeconomic backgrounds and geographies relevant to the study are represented in our 25-person team. This diversity supported our team's ability to internally question assumptions and discuss differences of opinion with a shared goal of better supporting a more collaborative, democratic research practice.

Cultivating this more collaborative dynamic was a conscious and continuous effort by the core team, in which some members benefit from more formal education, proficiency in English, and research experience to CBRs. Two core team members are, however, minority youth themselves and not all possess more years of professional experience to CBRs. While we may choose differently in future projects, terms like power were not often explicitly invoked in local language discussions with team members. As per explicit preferences of the group, we instead spoke in terms of "collaboration", sharing decision-making and leadership, and about practical concrete choices and behaviours that could democratise participation further. Local power structures of ethnicity, age, gender, socioeconomic status, education level, language,

**Table 1. Community-based researcher characteristics.**

| | Variable | Entire sample (n = 21) N (%) (* = national %) |
|---|---|---|
| Demographics | Mean age in years (range, S.D.) | 26.1 (23–29, ±1.7) |
| | Religion | |
| | Buddhist | 12 (57.1) (*70.1) |
| | Catholic | 2 (9.5) (*7.6) |
| | Hindu | 4 (19.0) (*12.6) |
| | Muslim | 3 (14.3) (*9.7) |
| | Ethnicity | |
| | Sinhalese | 13 (61.9) (*74.9) |
| | Tamil | 5 (23.8) (*11.2) |
| | Moor/Muslim | 3 (14.3) (*9.3) |
| | Gender | |
| | Woman | 12 (57.1) (*51.6) |
| | Man | 9 (42.9) (*48.4) |
| Marriage and family | Marital status | |
| | Single | 19 (90.5) |
| | Married | 2 (9.5) |
| | Living situation | |
| | Alone | 0 (0.0) |
| | Nuclear family | 17 (81.0) |
| | Extended family | 4 (19.0) |
| Socioeconomic factors | Highest education achieved | |
| | Vocation, NVQ, Dip | 2 (9.5) |
| | Degree or above | 19 (90.5) |
| | Current employment status[+] | |
| | Self-employed | 3 (14.3) |
| | Wage employed | 11 (52.4) |
| | Unemployed | 5 (23.8) |
| | Student | 2 (9.5) |

+ CBRs reported employment status beyond the PEER and current studies.

and regional location also required continuous attention. To that end, the core team took steps to maintain flatter power relations between themselves and CBRs and among CBRs through language and action including: (1) assigning language-matched debrief "buddies", avoiding terms like "line management" and "supervisory meetings"; (2) stressing open door policies; (3) privileging local languages and colleagues with less experience in different scenarios, and (3) organising team work in ways that would encourage all CBRs' voices to be heard. Debriefs were framed as "curiosity sessions" for both buddies, and intentionally designed to elicit feedback from CBRs on the process of doing the research, not just on the content being generated through it. The team leader (AP) aimed to position herself alongside and not above core colleagues by performing similar duties (e.g., debrief buddy) as much as possible. The core team did not seek to steer decision-making unless it was necessary given formal responsibilities.

## c. Data collection

SR reviewed study information with each CBR in language-matched written and oral forms; all provided written informed consent. As CBRs were employed by and reporting to the core

team, we took steps to minimise concerns over confidentiality, power, and performance, and reduce reporting or disclosure bias. First, CBRs pseudonymised submissions using a unique ID. Second, CBRs completed a brief, pseudonymised demographic form and received a structured journaling template in their preferred language. CBRs maintained personal cloud folders only accessible by SR until study completion. In line with guidance [18], no data were reviewed by core team members until the parent study and all CBR duties concluded.

Structured journaling was chosen over freewriting to support reflection across the team and project cycle on shared and underexplored issues of importance including CBRs' needs and experiences of conducting sensitive research [15]. As an autoethnographic method [27], journaling has been used by primary investigators as an aid for emotional release, process documentation, self-reflection, evaluation, and learning, but applied rarely with CBRs or in LMIC sensitive projects like ours [12, 18]. Following three PEER data collection rounds, CBRs answered four objective-orientated questions based on that specific round's experience. Questions were a guide and not intended to limit self-reflection. CBRs could type or handwrite entries in English, Sinhala, or Tamil. Professional translators supporting the larger project translated non-English entries; core team members cross-checked against original submissions for accuracy and reliability of English material pre-analysis. Translators were subject to confidentiality and data protection agreements. In total, 63 unique journal entries were submitted from the 21 CBRs.

Importantly, CBRs were also given research diaries for private and instruction-free use. In addition, debriefs, guided by a uniform protocol, were conducted as reflexivity sessions [2] between each CBR and an assigned core team member after every round. This may have encouraged self-reflection in multiple ways, potentially impacting journal entries, though CBRs understood these sources would not be subject to analyses for this study.

## d. Data analysis and co-production of this paper

Two core team members led application of Attride-Stirling's [28] six-step process for thematic analysis. AP first randomly selected 10% of journal entries for manual and largely inductive semantic coding, developing an initial coding framework (step 1). Journal entries were then uploaded in NVivo (QSR International) to support data management and ongoing analysis. SR tested the initial coding framework against a second 10% sample, proposing codebook adjustments as she identified themes (step 2). AP and SR double coded a third random 10% sample with minor revisions. ZR and KV independently reviewed the framework with minimal modification. Remaining data were coded under Basic Themes in the framework, adjusting labels as appropriate. Where common relationships were evident, Basic Themes were grouped to create Organising Themes. AP and SR constructed a thematic network (step 3), visually organising all existing themes in the software, (re)interrogating patterns and connections. Finally, four Global Themes capturing all mid- and lower-level categories supported more abstracted claims about CBRs' experiences in this study [28]. Results (steps 4–5) and their implications (step 6) are presented jointly in Results and Discussion and Recommendations. S1 Table provides selected quotes from one Global Theme illustrating early analysis to final themes.

As a measure of analytical quality control and member checking, all authors reviewed our results in all-team virtual workshops, jointly assessing the fairness of interpretations. Analysis was presented by theme and in local languages and used interactive mediums like Zoom and Miro to invite feedback from co-authors. CBRs had introductory thematic analysis training. While this was originally for inter-round PEER data analysis to support iterative tool development, CBRs were able to transfer analytical skills to assess this study's results and conclusions.

CBRs identified gaps which were rectified by the core team in subsequent drafts. A subset of CBRs reviewed the English draft in depth to support translation (RC, RD, TD, SK, LM, LS, PV) and all authors voted on a study title. The paper was approved by authors through a trilingual digital co-writing workshop. Trilingual 'how to' guides were developed for CBRs to fully participate in registering with and approving the final submission to the journal.

### e. Ethical considerations

Ethics approval was granted by University College London (REF 2744/007), the Institute for Health Policy (IRB/2020-026), and University of Colombo (EC-19-122). Limited demographic data were collected to preserve CBR privacy. Less than 2% of co-production literature comes from LMICs [2] and previous similar studies have not recognised CBRs' contributions as co-authors in research about them [15, 18], despite co-authorships' potential to strengthen CBR leadership, research translation and impact [14]. Co-authorship with our CBRs in this study is thus appropriate by editorial standards [29], and just, acknowledging proportional labour of Global South researchers–primary and novice [2, 30, 31]. We offer Sinhala and Tamil abstracts to aid local access to this study (S1 File). As CBRs are co-authors of this paper, quotes are pseudonymised [7]. Finally, while desirable in many instances, our dataset is not open access to ensure anonymity of PEER study participants who are (closely) socially connected to this study's authors and may be re-identified through its underpinning data. This is in keeping with standards for consent set forth by the ethics approval granted for this study. Additional information regarding the ethical, cultural, and scientific considerations specific to inclusivity in global research is included in the Supporting Information (S1 Text).

## 3. Results and discussion

We present four Global Themes encapsulating experiences of 1) learning, 2) personal-professional boundaries, 3) navigating challenges, and 4) support from the core team. Global and Organising Themes are summarised with quotes to aid interpretation. As Basic Themes were numerous, we do not illustrate them each in turn here, but present all themes in S2 Table for transparency.

### a. Learning about research, violence, and mental health

Learning experiences and (outstanding) needs across the project cycle received considerable attention from CBRs who highlighted professional and personal development and learning implications for their future contributions to community development.

**Skill and capacity development.** CBRs established, strengthened, or sought skills and professional capacities throughout the parent project. Like other peer studies [12, 18, 32], their on-the-job development was perceived as overwhelmingly positive including viewing COVID-19's role in changing original project plans as a learning opportunity: "Due to the pandemic, I got a chance to improve my knowledge. This was a great new experience for me. I learnt a lot and it was an opportunity to improve my skills" (ID14). CBRs identified improvements in interpersonal and interviewing skills, including practicing patience, "active listening skills" (ID03), question formation and ordering, "probing skills" (ID05) and managing emotionality and distractions. Our multi-round data collection cycle supported cumulative learning [32]: "I had an understanding about the things I should say and. . .not through the experience of the previous round" (ID21). Given changing data collection modes through the pandemic, CBRs necessarily gained key technological fluencies (e.g., video call platforms), honing observation skills of "body language and expressions" (ID06) and critical faculties such as noticing dynamics across in-person, video- or voice-based encounters, "giving more meaning to

[participants'] information" (ID06). Time management skills were tested in lengthier rounds: "As the questionnaire was long[er], more time was invested in this round. But I learnt how to manage my time" (ID19). Documentation skills and strategies using symbols, graphs and maps were developed to efficiently capture data and maximise flow during data collection. CBRs finally valued a multi-method approach, expanding their exposure to and comfort with diverse qualitative methods: "Story stem was a new and interesting exercise for both the participants and me. . . I was able to receive much information through Photovoice and using visual cards" (ID09). Despite this expansion of professional competencies, and preparation for incident management in line with responsible research practice [4, 7], some CBRs still desired greater confidence and skill to answer participants' help-seeking questions and attend to their personal reactions during data collection. This self-awareness indicates growing reflexive capacity–itself an essential practice within the qualitative paradigm [12, 32] which also supported personal growth, recognition of (sometimes blurred) role boundaries, and consideration for skill transferability for community action.

**Empathy and personal growth.** (Emotional) safety of researchers in sensitive studies is slowly gaining visibility [7], but seldom extends to CBRs [6, 33]. As an embodied experience, participatory violence and mental health research requires emotional connectedness and bearing witness to myriad forms of trauma with implications for wellbeing [7, 34, 35]. Our peer researchers identified ways their work impacted their internal worlds resonant with other LMIC-based CBRs [18]. First, as they acquired new knowledge of their communities through data collection, strong and at times negative emotions like shock, frustration, fear, and disgust were provoked [18, 36]: "I felt that I was shocked and frustrated when certain sensitive matters were discussed" (ID17). CBRs reported shifts in perspective due to peers' diverse and sometimes unexpected views [15], and humbly acknowledged both under- and over-estimating participants' and their knowledge: "Participants who I expected the least. . .from gave me very interesting content" (ID06). Rather than activating debate or open judgement, some CBRs reported that challenges to pre-existing personal assumptions prompted more "curiosity. . .to understand" (ID05). Like McCartan and colleagues [13], CBRs used the multi-round data collection cycles to re-evaluate personal standards and limitations: "[my debrief buddy]. . .made me realise. . .extraneous factors. . .cannot be controlled. . .[but] I can learn to minimise avoidable missed opportunities" (ID05). They also identified opportunities to apply learning to their personal lives, for example, "when [they] become a parent" (ID11). Importantly, no CBRs reported this emotional labour and learning as unreasonably difficult or unwelcome. Conversely, and as observed in other CBR projects [3, 18], application of expanded knowledge and skills developed self-confidence and feelings of empowerment: "It was a great experience for me. This hard time teaches me and shown me my capacity [sic]. I actually feel good" (ID20).

**Community-based researchers as community change agents.** Beyond personal growth, CBRs indicated two longer-term learning implications of benefit to their communities. First, as they participated in co-discovering their communities [18] and accumulated context-specific subject knowledge on violence and mental health, CBRs demonstrated aptitude for applying their intellectual and emotional learning towards identifying, appraising, and imagining more acceptable community responses:

As my participants and I engaged in a discussion about the existing support available, most of them were not affordable, accessible, and unable to handle incoming traffic. . . [future interventions] may entail considering the geographical location, vulnerabilities to accessibility, preferences of young victim-survivors of violence. . .training key personnel. . .to inculcate ethics, values, and practices to ensure responsible and accountable service delivery. (ID05)

CBRs refined their understanding of community needs across rounds. This individual capacity holds potential to support future efforts to identify and scrutinise context-specific solutions which may be more likely to result in community-level outcomes that matter [3, 19, 37].

Second, beyond possessing a strengthened aptitude for community development, several CBRs (re)evaluated their intentions, considering how best to take an active role in such efforts. In our setting, community-driven leadership is critical to overcome historical perceptions and disappointments of 'outsider' led initiatives. Our chosen PEER method encourages peer researchers' involvement through dissemination, in part to publicly position them as community focal points for future issue-based leadership [19, 25]. Many CBRs in our study expressed a desire to help raise awareness of project issues beyond their tenure, even if informally, and particularly around care-seeking options: "I can be a volunteer to spread awareness to these young women in the communities" (ID21). In a more formal capacity, several early career CBRs shared that project exposure to community conditions reinforced their enthusiasm for chosen occupations: "This made me feel passionate about my career. . .in social work because then I could help" (ID21). While the possibility for transformative social change within the local community and CBRs' role within it are both aspirations of participatory research [3, 14, 38, 39], some previous studies report feelings of disempowerment in the face of significant social suffering in low-resource settings [18]. Our CBRs instead appeared optimistic and change-orientated. While we cannot be certain, our PEER study's rare partnership model in which CBRs held key decision-making powers, participated from the project design phase, and have continued involvement through ongoing dissemination, may have fostered a greater sense of ownership and enduring capacity strengthening supportive of longer-term self- and community-actualisation [3].

## b. Personal-professional boundaries

The benefits and drawbacks of interviewing one's peers in research have been widely debated [3, 12, 37, 40–42]. CBRs' chosen parent-study participants ranged from very close friends to colleagues with whom socialising beyond the workplace had never occurred. CBRs then reflected upon their mixed experience conducting data collection within these social networks citing pros, cons, and the need for strategies to navigate their dual personal-professional roles.

Overall, and in line with other peer studies [12, 37], CBRs largely perceived quality of conversations and information were improved by greater familiarity–increasing transparency, feedback, and more open, detailed discussions: "I very well knew about two participants. . .not. . .much about the third person. The known participants shared information like speaking to a friend. The information I received from the third person was less" (ID12). Familiarity was particularly beneficial given the sensitive nature of the parent study: "Our pre-existing relationship was [a] help. . .they were open. . .because they trust me" (ID20). CBRs also felt prior relationships supported participants' "willingness to [commit and] engage" (ID02) in three rounds of data collection over an extended period without prior "monetary gains" (ID02) or other incentives.

In contrast, some researchers identified disadvantages to personal relationships including a sense that participants sometimes held information back due to known common peers and/or presuming CBRs already knew certain facts or stories: 'They were not inclined to talk about things deeply, they were not taking the interview seriously and sometimes they just assumed that I knew some of the facts and their opinions' (ID21). Mutual familiarity with community members "already known" (ID20) to CBRs also meant participants sometimes 'overshared' and had to be encouraged to maintain anonymity of others: "I had to say, 'don't mention

names'" (ID20). Beyond over- or under-sharing information, some participants became easily distracted and "went off the track" (ID03), diverting conversations away from the study's focus–a difficulty observed in other peer studies perhaps reflecting the atypical nature of interview-style discourse for both parties [37]. This sense of informality made it difficult to establish boundaries with (some) participants; a challenge requiring a balancing act across personal-professional roles that can be difficult to maintain [41, 43, 44].

To aid this balancing, CBRs developed and practiced four strategies. First, they made conscious decisions about participant selection, drawing on core team guidance around eligibility, inclusivity and suitability for research participation [17]. Many researchers intentionally chose not to interview peers overly similar or "very close to [them]" (ID17), believing some distance may strengthen their professional identity and legitimacy and enable them to "conduct a professional and successful interview" (ID17) while simultaneously offering new, and diverse views [19]. Next, and when possible given COVID-19 and technology conditions, CBRs encouraged in-person or at least video-call modes to support connectedness during data collection–a beneficial practice for cultivating intimacy and supported engagement for sensitive research and during the pandemic [45, 46]. Third, CBRs used clear communication of expectations and appropriate disclosure practices *prior* to interviews to establish boundaries and a sense of professionalism from the outset:

> I decided to approach and communicate with my peers/friends with professionalism to help establish a clear boundary between data collector and friend/peer. For example, scheduling a date and time for an interview would entail a call to my participants followed by an email with relevant information. . . advising them to take a look at these documents prior to the meeting, so that I can address any questions before the interview. (ID05)

Finally, CBRs had to actively reinforce boundaries during the interview process, for example encouraging that all casual conversation occur after the interviews and steering conversations "back to the main topic" (ID03) when required. Boundaries were perceived to be clearer and easier to maintain by later rounds of data collection due to both parties' practice and CBRs' reinforcement [19]. In the next section, we extend our discussion of boundaries amongst other challenges, and explore CBRs' resourceful efforts to overcome them.

### c. Navigating challenges

We identified four primary challenges in conducting sensitive participatory research in our context including interpersonal, time management, health and safety, and technological difficulties, with COVID-19 producing cross-cutting impacts. We explore their impact upon and mitigation by CBRs in turn below.

### Identified challenges

Interpersonal challenges, including expanded reflection upon boundaries, took centre stage in CBR journals. Like other peer studies [18], participants' busy schedules presented hurdles for completing data collection, creating atypical working hours: "I had to conduct late night interviews because my participants weren't available during the daytime. . .it was a bit difficult for me" (ID13). At the extreme, managing interview bystanders required multiple rounds of rescheduling, for example when participants joined during work hours and discontinued to attend to colleagues or clients. Some CBRs reported participant reticence speaking about sensitive issues, such as "sexuality and [pre-marital love] affairs" (ID17), when family were home.

Distractions and interruptions also arose in the form of phone calls, text messages and food deliveries to participants–the latter a lifestyle impact of COVID-19 in our setting.

Participants' and sometimes CBRs' emotions were, however, the most discussed interpersonal challenge, as observed in other research groups [16, 33]. Discomfort with video calls in certain circumstances, participants' confusion and fear of questions due to perceived inadequate knowledge, both parties' stress around scheduling and managing technological difficulties were all reported, in addition to initial nervousness, perceived lack of control, and frustration for CBRs. Given their youth demographic, some participants balancing employment and higher education seemed over-stretched during interviews: "[She was] extremely exhausted. . .due to university assignments therefore I did not receive the maximum amount of information from her" (ID06). Although PEER seeks third-person perspectives [25], two participants became "emotional" (ID19) self-disclosing personal stories, one pausing the interview temporarily, only continuing "after discussing things" (ID14) with their friend and CBR. Notably, managing emotions was not regarded as concerning for CBRs as other challenges raised below. Despite varied options for onward support, in line with best practice [7, 12], neither participants nor CBRs disclosed utilising them nor requested assistance from the core team, and no serious adverse events were reported. This may ease funder and ethics board anxieties around 'risk' in sensitive research [34], but likely points to our proactive supportive working conditions mitigating concerns as presented later.

In addition to scheduling challenges, time management proved challenging for some CBRs and participants, particularly in lengthier rounds with multiple methods. Breaking interviews into multiple sessions was sometimes necessary. Rare, but important, CBRs' personal health impacted timelines, creating additional stress: "It was really hard for me. I have [a chronic health condition] as well. That was a real stress time period for me [sic]. These days, I haven't enough sleep" (ID20). One CBR and participant were hospitalised, while one core team member and one CBR suffered family bereavement for non-COVID reasons mid-project, all requiring support. Technology, chiefly internet connectivity, presented regular difficulty for all CBRs during virtual data collection [47]. Extreme weather in monsoon season, power cuts, and poor local infrastructure in less connected areas sometimes rendered video-calls too difficult. CBRs felt this reduced their ability to engage with "participants' emotions and body reactions. . .via phone" (ID07), impacting interview quality.

A cross-cutting challenge, COVID-19 impacted our research practice in every way [48], extending the entire project timeline, altering fieldwork, increasing technological challenges, and affecting the health and safety and socioeconomic circumstances of all parties. Unlike older generations' careers affected by the country's civil war, COVID was the first significant disruptor impacting our CBRs' early working lives, testing their resilience. Akin to other peer studies in COVID [46], our CBRs navigated shifting data collection to phone, WhatsApp, or Zoom as government COVID-19 responses introduced risk-reduction efforts: "During the second interview the Covid-19 pandemic situation worsened. Travel restrictions were imposed. Therefore, the interviews were not done face-to-face" (ID21). Pandemic-induced shop closures and travel constraints meant photocopying materials and reaching open establishments became difficult, while postal disruptions made sending supplies difficult. In more mobile windows, face-to-face interviews were preferred, but introduced interpersonal challenges of conducting interviews in participants' homes with more bystanders and less privacy than usual. Most significantly, COVID-19 directly affected the families, health, and wellbeing of all parties. Family and multiple personal infections required one core team member, one CBR, and one participant to quarantine at home, while another participant was quarantined at a mandated government-run centre: "My participants were victims of the COVID-19 virus. Even I had to go through quarantine and had to face difficulties continuing with the interviews" (ID08).

Tragically, one CBR and two participants lost family members to COVID-19 mid-project. Remarkably all our affected participants and CBRs chose to continue, even despite one team's focus on COVID's impact on youth mental health and their participants' direct experiences of loss. While discontinuation without penalty was offered, in the spirit of true co-production, we respected CBRs' and participants' agency to choose involvement for themselves [33].

### Strategies for navigating challenges

CBRs developed communication, planning, and technology strategies to (partially) overcome challenges encountered during fieldwork [14, 47], applying learning from training and previous rounds [19].

Efforts to reduce social desirability bias [49] saw CBRs deploy communication strategies including asking participants to move location in the home to avoid distractions and interruptions from bystanders. Engaging with bystanders directly also helped to agree a manageable way forward, as practiced in foundational training: "At the training session, the core team created a role play scenario to help us deal with interruptions from family members. The activity was helpful in managing the challenge" (ID05). Insightfully, some CBRs asked participants about vocabulary preferences to discuss culturally sensitive topics like "sex and violence" (ID06) and sought private settings away from family bystanders to discuss them: "I asked. . .if he needs to move to a free safe place. He agreed and we went outside to his garden. . .after that, he spoke more openly and comfortably" (ID17). During interviews, CBRs "gave a short break" (ID07) to ease emotionality, simplified questions and granted extra time for participants expressing confusion or fear over their perceived 'lack of subject knowledge'. Some CBRs ended interviews by "check[ing] in with the participants to make sure they [had] a chance to express their opinions and feeling about the interview" (ID05)–something CBRs deemed a worthwhile practice. Finally, honest and timely communication with the core team was used to navigate delays and setbacks.

Pre-paid allowances for internet expenses for CBRs and participants removed cost hurdles to accessing devices and data and demonstrated trust in CBRs: "Even though there were technical difficulties, the participants were encouraged by giving them expenses for internet facilities" (ID02). However, challenges with internet connectivity and power required compromises like sharing visual tools via WhatsApp as opposed to screen sharing in real time and choosing when during interviews to use video- over voice-based calls: "Since the conversation about violence and harm can be triggering to some, I asked the participant to turn on their camera, to monitor their verbal and non-verbal behaviour" (ID05).

Planning strategies demonstrated initiative and anticipated interpersonal and technology challenges. CBRs used planning to minimise challenges in subsequent rounds by drawing on successful strategies of previous rounds and other CBRs, as exemplified below:

> I made sure to communicate. . . what is expected of [participants] during the interview. As this round was occurring through Zoom, I made sure that participants understood the importance of 1) situating themselves in a space with privacy and little interruption from others. . ., 2) having the video on. . ., and 3) having sufficient data to ensure smooth communication. . . participants were informed that this round will be lengthy compared to the last round. . . Communicating the above at least a week before the interview was helpful in preparing the participant for Round 2. (ID05)

### d. Experiences of (support from) the core team

Our fourth and final global theme, CBRs reflected on the role of the core team and the nature of support received by them across the project cycle. Their reflections focused on working and

interpersonal conditions they valued, and the particular gestures or provisions offered by the core team believed to support these conditions; we review them in turn below.

CBRs vocalised an overall appreciation for the core team and felt able and enabled to reach out for assistance throughout the project [14], often multiple times: "The support received from the core team was highly beneficial in how I was able to manage this challenge" (ID05). Unlike some sensitive studies, including those with primary investigators and in highly resourced settings [44], our CBRs did *not* raise concerns around feeling isolated or unsupported. Many instead noted sufficient support due to core team accessibility and responsive communication: "As the main team maintained a good communication with the interviewers, I could ask about anything, at any time" (ID21). The core team was perceived as "understanding of the situation" (ID05), flexible and willing to negotiate reasonable changes in light of CBRs' own schedules, considering their role in the parent project was not always their sole commitment, and accommodating of participant challenges. CBRs also noticed the core team's anticipation of their needs, particularly with pre-fieldwork training, illustrated later in this section [7, 50].

Five key core team provisions, or gestures, were identified as contributory to positive interpersonal dynamics and ultimately CBR performance. First, accessibility and consistent, comprehensive communication was buoyed by provision of multiple communication modes and the core team's ability to communicate in all languages, supporting a more inclusive team: "Any method (WhatsApp, call, text) convenient to us could be used to contact them" (ID21). Second, training programmes were perceived to prepare CBRs pre-fieldwork and render new challenges more manageable [7]: "Although it was a new experience, it was not that difficult because of the training" (ID19). Some researchers expressed confidence to manage challenges (e.g., disrupted interviews) on their own, due to the trainings and component activities like role play, later reducing their need to request additional help. As in other peer studies [12, 13], initial training provided a place to comfortably raise pre-fieldwork concerns, while debriefs–a third provision between rounds–continued this protected sharing space. CBRs also valued debriefs for facilitating self-reflection of their own limitations during fieldwork [39]. Coupled with a fourth provision of extra team or one-to-one practice sessions with core team members, CBRs felt better able to navigate the research process and work towards improving over subsequent rounds. This multi-modal support is well-illustrated below:

> . . .after the two [initial] days of training, I personally did not feel ready to conduct the interviews. I felt that I needed more practice. Maybe the core team felt this lack of readiness. . .- which led them to hold [additional] training. . . it helped build my confidence as a data collector. We were given an opportunity and a space to practice, and. . .feedback as to what we could have done differently, and what best practices we could continue to do. This feedback and [later] practice sessions helped me develop direction and intuition as a data collector. Furthermore, it certainly shaped my ability to deal with the stuckness [sic] that I experienced during interviews. (ID05)

Finally, resources both in terms of time and consumable inputs such as internet and travel allowance, further support brochures, IT equipment, and end-of-project vouchers as tokens of appreciation for participants, were appreciated by CBRs and participants alike, noting that "with them, the participants became confident and trusting about the interviews" (ID21). In particular, technology like phones, cameras and laptops, and associated running costs were critical resources for remote participation. All CBRs had access to basic smart phones and internet connection and were furnished with data in advance so as not to disadvantage any individual's access to internet throughout the study. Other inequalities in technology access

were addressed through offers to provide devices to CBRs according to their needs to enable fair participation. Some CBRs chose not to accept additional devices (e.g., laptops), preferring to work in other mediums, and the core team respected these decisions as it did not impact CBRs' ability to join discussions nor to deliver quality work. Other projects may require more uniform use of technologies. These resources, emphasising wellbeing, were particularly critical during COVID-19 in recognition of the "mental and emotional toll of the pandemic in their personal lives" [46, p7].

## 4. Recommendations and conclusion

As an extension of our learning and to address current shortcomings in the co-production evidence base [2], we collectively derived a set of concrete recommendations for our future selves and others considering sensitive participatory research. These recommendations reflect a compilation of intentional choices about project design, resourcing, and actions we would repeat, improve upon, or introduce into our own practice based on this study's data and additional co-writing workshops for this paper. Table 2 presents a checklist of recommendations in five priority areas: Team communication culture; Project design; Foundational training; Human and consumable resources; and Considerations for volatile research contexts. S3 Table offers an expanded checklist with specific examples from our project experience. Noting that both this and the parent study were low resource and conducted during the COVID-19 pandemic, we believe our recommendations are reasonable and feasible for most methodologically similar projects. We offer these with humility, recognising our own need for adaptive learning, and privileging the often under-valued and sometimes alternative or even opposing insights of CBRs [31]. We hope these contribute to evolving and necessary conversations in global health and development around decolonisation and formation of more equitable, wellbeing-focused research partnerships, particularly those in LMICs and focused on sensitive topics like violence and/or mental health [17, 20].

Many recommendations concern all team members, while some are necessarily aimed at Principal Investigators, other senior researchers and even funding bodies with whom decision-making power disproportionately lies, to set the tone for projects from the outset [2, 7]– including pre-recruitment and as early as the Conceptualisation Phase of projects when problems, partners and funding are first being selected and secured to explore global health issues. Global health organisations based in the Global North benefit from and create privilege by affiliation for local collaborators in a field that remains inequitable [51]. Such institutions hold considerable power at the Conceptualisation Phase. Acknowledging the dynamic that our project's funder and partners add to our study, we build on calls to act as "allies and enablers to local processes and learning" rather than as agenda-setters [51, p1628]. As matters of power are rarely explicitly addressed in global health and development including community-based and violence studies [2, 17], we emphasise items (*) early career and community-based researchers could enact independently–even at a task level and despite otherwise under-supportive conditions. We do so not with a view to place undue responsibility upon junior colleagues but conversely to foster a sense of empowerment and control over key choices.

To aid readers further, we modified our checklist to provide a visual Opportunity Timeline for when teams may find it most advantageous to identify a need, budget for, and/or (plan to) deliver particular recommendations, grouped by our five priority areas (Fig 1). We developed a broad phasing for global health projects applicable to most research and implementation teams: Conceptualisation constitutes the establishment of the project-partnership-funding triad, followed by Preparation (i.e., project design), Implementation (i.e., data collection, analysis, trials and other 'doing' activities), and Dissemination phases. We recognise projects rarely

**Table 2. Recommendations checklist for supportive team dynamics in sensitive participatory research in low-resource settings.**

| Priority Area |
| --- |
| *Team Communication Culture* **(C1-C7)** |
| C1. Explicitly communicate openness to alternative and/or dissenting views with (junior) staff. |
| C2. Agree clear team roles, responsibilities, and accountability for all regardless of seniority. |
| C3. Agree comportment standards for all including zero-tolerance policies if appropriate. |
| C4. Pair CBRs with a consistent, (language) accessible core team member for project duration. |
| C5. Confirm team communication mechanisms and methods and when each should be used. |
| C6. Core team members to maintain multi-channel availability for project duration. |
| C7. Normalise respectful, consistent, responsive feedback practices amongst all colleagues.* |
| *Project Design* **(D1-D7)** |
| D1. Structure data collection phase to allow repeat opportunities for CBRs to apply, test, and improve data collection skills. |
| D2. Build-in individual-* and team-based reflection and review processes during and post-project. |
| D3. Anticipate and support additional practice sessions for CBRs.* |
| D4. Create 'choice opportunities' that foster CBR decision-making throughout project. |
| D5. Embrace, model and expect flexibility when it will benefit team wellbeing and/or evidence.* |
| D6. Design mechanisms for preventing and managing adverse events during fieldwork. |
| D7. Plan resources for CBRs to design and lead dissemination activities; visibilise them as knowledgeable change agents in their communities [2], if desired and safe. |
| *Foundational Training(s)* **(T1-T7)** |
| T1. Support emotional safety in sensitive research [7] by introducing evidence-informed strategies for managing discomfort, reflexivity, and personal beliefs being challenged and/or changed.* |
| T2. Create interactive, multi-method opportunities to practice navigating diverse interpersonal dynamics including challenges and de-escalation.* |
| T3. Practice and receive feedback on data collection skills, particularly given data collection moments may be re- or vicariously-traumatising for participants or CBRs in sensitive research [7]. |
| Discuss and co-develop 'further support' materials *with* CBRs to build their confidence to assist participants and themselves; pre-made materials without in-depth review may leave CBRs under-prepared in situ. |
| T4. Reenforce that CBRs should not (be expected to) compromise their own safety and wellbeing to accommodate participants or other project needs [7].* |
| T5. Practice general professional skills e.g., documentation practices, time and expectation management. |
| T6. Trial project technologies and tools including trouble shooting strategies. Do not presume all parties possess requisite items/devices nor are competent and comfortable (independently) applying them–practice how they should be used within the project*. |
| *Human and Consumable Resources* **(R1-R9)** |
| R1. Dedicate strategic time and effort to identify, recruit and retain suitable CBRs. |
| R2. Consider exercises to identify strong team leaders amongst CBRs early on as they may motivate other CBRs differently to the core team. |
| R3. Build-in extra time to allow for project slippage due to unforeseen challenges.* |
| R4. Ensure necessary and multiple technologies are available for all, particularly for remote data collection methods; offer alternatives and/or catch-up time if primary technologies malfunction. |
| R5. Provide project tools in multiple and accessible formats. |
| Provide reasonable allowances for cost-incurring activities for CBRs *and* participants. |
| If appropriate, co-agree end-of-project gifts of thanks and compensation for participants *with* CBRs. |
| R6. As appropriate, offer gestures recognising CBRs during and post-service. |
| R7. Offer wellbeing resources for all parties–particularly in violence and mental health projects [7]. |
| *Considerations for Volatile Research Conditions†* **(V1-V9)** |
| In addition to tackling sensitive research, teams may face other and changeable working conditions due to natural or human-made disasters, (civil) unrest and conflict, health crises, and more. Drawing on our experience navigating the COVID-19 pandemic and parallel economic crisis, we propose: |

*(Continued)*

**Table 2.** (Continued)

| |
|---|
| V1. Identifying potential challenging scenarios and project 'threat' levels with CBRs to anticipate (un)likely conditions.* |
| V2. Developing mitigation strategies with high, medium and low levels of control for CBRs to consider how they could adapt to changing conditions.* |
| V3. Jointly discussing how CBRs would feel in each scenario, what and how best to deploy strategies* with appropriate inputs from the core team. |
| V4. Formalising and reviewing safety plans, including extraction planning. |
| V5. Ensuring appropriate insurance is in place for field-based team members. |
| V6. Providing project-tailored health and safety kits for fieldwork/travel. |
| V7. Collectively anticipating and accepting undesirable shifts in the project may be necessary.* |
| V8. Openly discussing the potentiality for project interruption and/or cessation in extreme circumstances and reiterating no research is more important than wellbeing.* |
| V9. Transparently agreeing employment conditions and commitments under such circumstances to foster mutual trust. |

* Denotes recommendations which could be (partially) independently enacted by CBRs.

† While we stress these recommendations for volatile contexts, all items are relevant and reasonable for projects deemed low-risk at their inception as field conditions can and do change swiftly.

follow linear and strictly delineated phases and that our Opportunity Timeline is necessarily an oversimplification of our experience combined with our learning of what we would like to see and practice in future projects.

We also indicate where recommendations can span multiple phases. We do so firstly to indicate when consideration and accountability could start, including before more junior colleagues tasked with delivering global health projects are typically hired. Principal investigators, funders and ethics bodies ideating and identifying funds may consider the need for e.g., certain budget lines, safety protocols, or buffers in time and resources, especially for work in volatile contexts, from Conceptualisation. This can reduce risks for future colleagues, including CBRs, of unreasonable or even unsafe conditions which may be difficult for junior parties to change. Second, straddled phasing indicates there are multiple opportunities to revisit a recommendation, although its implementation may change in form or function over time. Finally, there are recommendations we believe are truly cross-cutting. Leadership may begin from senior colleagues and evolve throughout the project with core teams and CBRs working together to sustain a recommendation in the most suitable applied form.

Both our checklist and opportunity timeline offer tools for wellbeing- and growth-minded researchers of all levels to advocate for improved project conditions with research funders, employers, ethics bodies and partners and calls on these latter actors to proactively provide for researchers in all roles.

This is the first study to explore the experiences of CBRs in a sensitive co-produced research project from Sri Lanka and one of the few to do so from a LMIC and global health discipline [2, 3]. Only 21 CBRs from one national project participated, and we could not assess long-term learning retention nor their subsequent community action. However, CBRs' immediate post-project lives have included securing internships and employment in relevant non-profits, and places on international and issue-related education courses with core team support. CBRs continue to provide analytical leadership, producing creative outputs through extended employment. Our CBRs' cumulative reflections offer insight into the multiple and feasible opportunities diverse global health actors can *create* to foster curiosity, personal and professional development, and wellbeing within research teams [31].

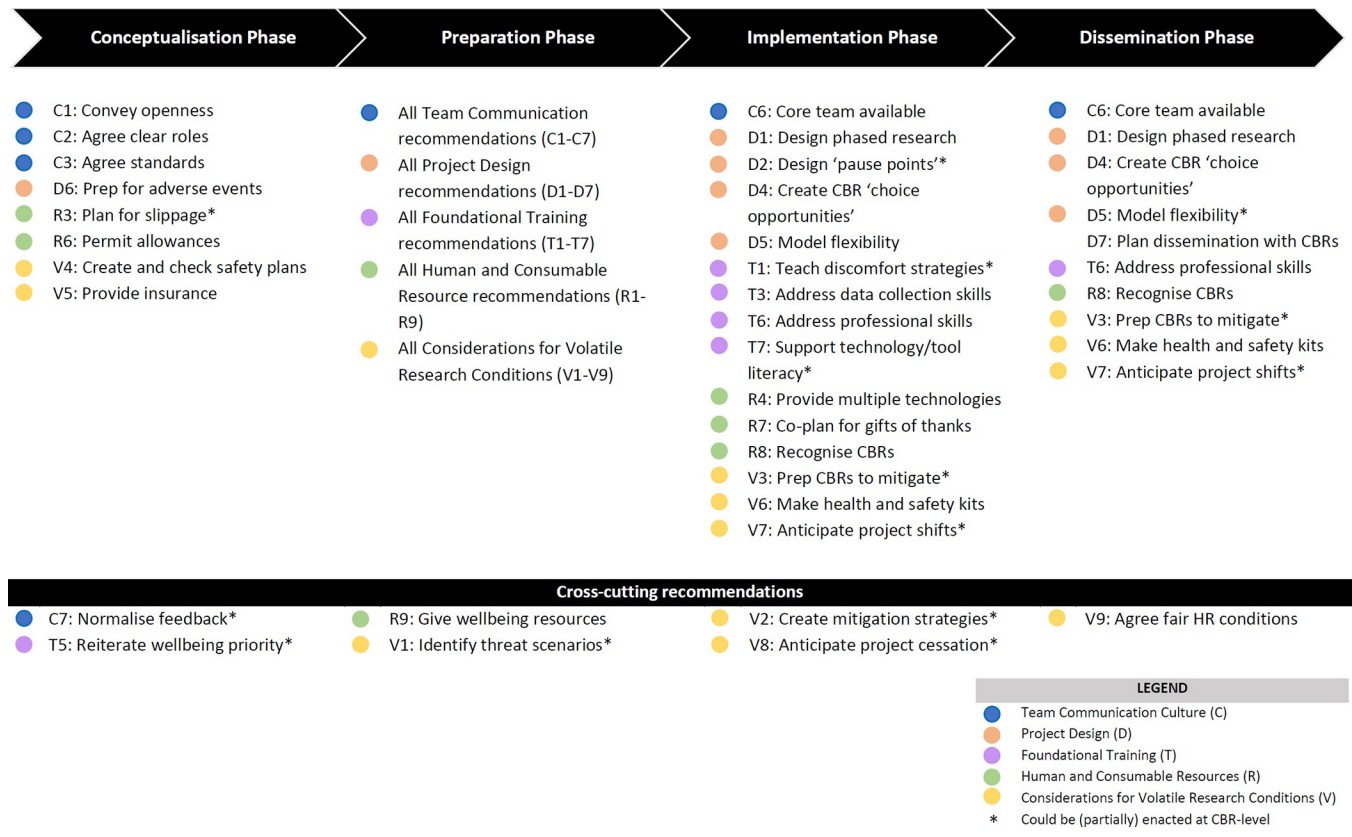

**Fig 1. Opportunity timeline for recommended actions to support wellbeing and development of community-based researchers (CBRs).**

Contrary to certain anxieties amongst research actors, emotional and physical safety did not emerge as an area of significant concern for CBRs in this violence and mental health project [7]. However, we believe this was by design, not happenstance, and should not be taken to suggest (emotional) safety in sensitive research is unimportant. Far from it, we believe professional researchers, their host institutions, funders, and ethics boards should work towards strengthened minimum standards for project conditions prioritising participant, CBR, and senior researcher welfare on equal footing. Research actors should avoid adopting an over-corrective risk aversion to sensitive projects, but must recognise that failing to prevent burnout or other harms in sensitive projects risks losing talent altogether, undermining progress on the very issues for which institutions claim to care [7, 50].

Further, particular attention to quality of communication, training, human and consumable resourcing, project structure, and considerations for volatile research contexts, "offer[s] the opportunity to change the [team] dynamics and strengthen co-researcher capacities, both within the co-production process and in their targets of change" [2, p2] beyond the boundaries of the project. As participatory methods are increasingly deployed to research pressing and potentially high(er)-risk issues like violence and mental health [15], collaborative and reciprocally generative relationships among professional researchers, CBRs, participants and all other stakeholders are critically valuable, but require committed, intentional and reflexive research design and co-implementation to become practically possible.

## Supporting information

**S1 Table. Theme 1 example data analysis.**
(DOCX)

**S2 Table. Themes with organising description.**
(DOCX)

**S3 Table. Full recommendations with examples.**
(DOCX)

**S1 File. Sinhala and Tamil.**
(DOCX)

**S1 Text. Questionnaire on inclusivity in global research.**
(DOCX)

## Acknowledgments

We extend our gratitude to study participants for their significant contributions and perseverance despite challenging conditions. Secondly, we thank our consortium colleagues and administrative and finance partner (Institute for Health Policy) for supporting this work.

## Author Contributions

**Conceptualization:** Alexis Palfreyman.

**Formal analysis:** Alexis Palfreyman, Safiya Riyaz.

**Investigation:** Alexis Palfreyman, Safiya Riyaz, Zahrah Rizwan, Kavitha Vijayaraj, I. P. R. Chathuranga, Ruwanka Daluwatte, W. A. T. Devindi, B. Shakila Eranda, Vinodani Jayalath, Azam Junaid, Ashra Kamal, Shehan Koshila Kannangara, K. M. G. Prasanga Madushani, Laksi Mathanakumar, S. Ihjas Mubarak, Vithusha Nagalingam, Sachin Palihawadana, Ravishanka Pathirana, V. G. Sameera Sampath, Lojini Shanmuganathan, Tharindi Thrimawithana, Priyatharshiny Vijayaratnam, Sasith Lakshan Vithanage, R. K. K. A. Sajini Wathsala, R. Mervin Yalini.

**Methodology:** Alexis Palfreyman, Safiya Riyaz, Zahrah Rizwan, Kavitha Vijayaraj, I. P. R. Chathuranga, Ruwanka Daluwatte, W. A. T. Devindi, B. Shakila Eranda, Vinodani Jayalath, Azam Junaid, Ashra Kamal, Shehan Koshila Kannangara, K. M. G. Prasanga Madushani, Laksi Mathanakumar, S. Ihjas Mubarak, Vithusha Nagalingam, Sachin Palihawadana, Ravishanka Pathirana, V. G. Sameera Sampath, Lojini Shanmuganathan, Tharindi Thrimawithana, Priyatharshiny Vijayaratnam, Sasith Lakshan Vithanage, R. K. K. A. Sajini Wathsala, R. Mervin Yalini.

**Project administration:** Alexis Palfreyman, Safiya Riyaz.

**Resources:** Alexis Palfreyman, Safiya Riyaz.

**Software:** Alexis Palfreyman, Safiya Riyaz.

**Supervision:** Alexis Palfreyman, Safiya Riyaz, Zahrah Rizwan, Kavitha Vijayaraj.

**Validation:** Alexis Palfreyman, Safiya Riyaz, Zahrah Rizwan, Kavitha Vijayaraj, I. P. R. Chathuranga, Ruwanka Daluwatte, W. A. T. Devindi, B. Shakila Eranda, Vinodani Jayalath, Azam Junaid, Ashra Kamal, Shehan Koshila Kannangara, K. M. G. Prasanga Madushani, Laksi Mathanakumar, S. Ihjas Mubarak, Vithusha Nagalingam, Sachin Palihawadana,

Ravishanka Pathirana, V. G. Sameera Sampath, Lojini Shanmuganathan, Tharindi Thrimawithana, Priyatharshiny Vijayaratnam, Sasith Lakshan Vithanage, R. K. K. A. Sajini Wathsala, R. Mervin Yalini.

**Visualization:** Alexis Palfreyman, Ruwanka Daluwatte.

**Writing – original draft:** Alexis Palfreyman, Safiya Riyaz, I. P. R. Chathuranga, Ruwanka Daluwatte, W. A. T. Devindi, Shehan Koshila Kannangara, Laksi Mathanakumar, S. Ihjas Mubarak, Lojini Shanmuganathan, Tharindi Thrimawithana, Priyatharshiny Vijayaratnam.

**Writing – review & editing:** Alexis Palfreyman, Safiya Riyaz, Zahrah Rizwan, Kavitha Vijayaraj, I. P. R. Chathuranga, Ruwanka Daluwatte, W. A. T. Devindi, B. Shakila Eranda, Vinodani Jayalath, Azam Junaid, Ashra Kamal, Shehan Koshila Kannangara, K. M. G. Prasanga Madushani, Laksi Mathanakumar, S. Ihjas Mubarak, Vithusha Nagalingam, Sachin Palihawadana, Ravishanka Pathirana, V. G. Sameera Sampath, Lojini Shanmuganathan, Tharindi Thrimawithana, Priyatharshiny Vijayaratnam, Sasith Lakshan Vithanage, R. K. K. A. Sajini Wathsala, R. Mervin Yalini.

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
