## [Decision Letter · Decision Letter 0]

29 Apr 2022

PGPH-D-22-00472

Cultivating capacities in community-based researchers in low-resource settings: Lessons from a participatory study on violence and mental health in Sri Lanka

Dear Dr. Palfreyman,

Thank you for submitting your manuscript to PLOS Global Public Health. After careful consideration, we feel that it has merit but does not fully meet PLOS Global Public Health’s publication criteria as it currently stands. Therefore, we invite you to submit a revised version of the manuscript that addresses the points raised during the review process.

Note important comments especially from reviewer 2 which needs extensive editing, additional details and redrafting of several sections of the paper, especially the gaps pointed out in the Methods. 

Please submit your revised manuscript by . If you will need more time than this to complete your revisions, please reply to this message or contact the journal office at globalpubhealth@plos.org. Please include the following items when submitting your revised manuscript:

We look forward to receiving your revised manuscript.

Kind regards,

Prashanth Nuggehalli Srinivas, MBBS, MPH, PhD

Academic Editor

Journal Requirements:

2. Your manuscript is missing the following sections: Results. Please ensure these are present, and in the correct order, and that any references to subheadings in your main text are correct. An outline of the required sections can be consulted in our submission guidelines here:

https://journals.plos.org/globalpublichealth/s/submission-guidelines#loc-parts-of-a-submission

3. In the online submission form, you indicated that "Our dataset is not open access to ensure anonymity of socially-connected study participants who may be re-identified through this study’s data and authorship.". All PLOS journals now require all data underlying the findings described in their manuscript to be freely available to other researchers, either 1. In a public repository, 2. Within the manuscript itself, or 3. Uploaded as supplementary information.

4. We have noticed that you have uploaded Supporting Information files, but you have not included a list of legends. Please add a full list of legends for your Supporting Information files after the references list. 

Additional Editor Comments (if provided):

Reviewers' comments:

Reviewer's Responses to Questions

**Comments to the Author**

1. Does this manuscript meet PLOS Global Public Health’s publication criteria? Is the manuscript technically sound, and do the data support the conclusions? The manuscript must describe methodologically and ethically rigorous research with conclusions that are appropriately drawn based on the data presented.

Reviewer #1: Yes

Reviewer #2: Yes

2. Has the statistical analysis been performed appropriately and rigorously?

Reviewer #1: N/A

Reviewer #2: N/A

3. Have the authors made all data underlying the findings in their manuscript fully available (please refer to the Data Availability Statement at the start of the manuscript PDF file)?

Reviewer #1: No

Reviewer #2: No

4. Is the manuscript presented in an intelligible fashion and written in standard English?

Reviewer #1: Yes

Reviewer #2: Yes

5. Review Comments to the Author

Reviewer #1: This manuscript reports learning from a case study in which community-based researchers contributed significantly to particularly sensitive mental health research conducted in Sri Lanka. Employing community-based researchers is a growing practice and has potentially immense benefits for research outcomes and impact, the community-based researchers themselves. The strategy also comes with risks, which need to be mitigated and managed strategically from the outset. This manuscript presents a rigorous and thoughtful leap towards doing so and is likely to be hugely appreciated by research teams seeking to employ these methods safely and ethically, in LMIC and other settings. I have only three suggestions for the authors to consider.

First, some funders and publishers require datasets to be open access. Although there can be clear ethical and other reasons to argue against this in individual cases, it is a pity if a solution to this cannot be found in similar circumstances as the current case study presents. I encourage the authors to consider including in their recommendations ways to allow similar data sets to be open access, if at all possible, and to demonstrate one solution for the data set on which the manuscript is based.

Second, Table 1 (Community-131 based researcher characteristics) might benefit from an additional column providing whole-country demographic information on the relevant dimensions for comparison purposes.

Third, the analysis is rigorously conducted and evidenced and provides an excellent basis for the recommendations articulated. However, it is presented as a list. My experience is that usually themes/sub-theme analysis can be taken to the next level of analysis in the form of a model. At this next level, inter-relationships between themes and sub-themes are identified, possibly with feedback loops and posited intervention points. If the authors would like me to share an example I am happy to do so privately. I am thinking a particular study (currently in re-review) in which I was encourage by my collaborators to take a list of themes to the next level of integration which allow us to have a much clearer view of the processes involved in implementing change.

Reviewer #2: Comments

1. Overall commend the concept of this study – important to examine and consider this topic and interesting methods and reflexivity – thank you

2. Abstract - English needs further editing – some of the sentences had to be read several times to understand – an edit to get a brisk and clear meaning recommended – eg. This sentence in abstract "Participatory methods, which rely heavily on community-based data collectors, are growing in popularity to deliver much-needed evidence on violence and mental health in low- and middle-income countries facing the greatest burden, but fewest resources to respond".

3. Introduction – Sets the scene – however the sequence of paragraphs could be reviewed e.g. in first paragraph the objective of study is stated – (lines 42 -44) and this is then replicated with expansion at end of Introduction section - generally feel that it could be more concise with less repetition.

4. Introduction - While in 36 – 38 the authors outline growing popularity of participatory methods – they have not engaged in any depth with the reasons for its’ importance, and have not engaged with some key discussions around decolonisation in global health/ the power relations in community coproduction of knowledge eg Abimbola S. The uses of knowledge in global health. BMJ Specialist Journals; 2021. Or eg 2 Abimbola S, Pai M. Will global health survive its decolonisation? Lancet. 2020;396(10263):1627-8.

5. Methods – given the challenges of including CBRs in research question design – it would be useful to describe how or whether CBRs were involved in writing the grant for this research – and if they were – how they participated – and if they weren’t, how the grant proposal gave space for new question development. Please elaborate as the inception of this study is relevant to the power relations and peer researcher proposal and to understand the primary investigator/ core team roles vs the CBR roles.

6. Methods - Stance and reflexivity – core team/ CBR – while this paper includes many aspects of reflexivity, it would be useful to understand the privilege/ background of authors and how the core team vs CBR relationships were managed to minimise hierarchical power relations – please add this to the Methods section

7. Methods - Recruitment of CBR – given the impressive representativeness and diversity – it would be good to understand further detail on how the CBR were recruited.

8. Findings/ discussion – the four themes are relevant and informative and discussion of each theme engages well with relevant literature. It would be useful to include further discussion of the ways that technology/ access to technology influenced the research process and contributions e.g. were there ways that more remote CBRs could not use their video’s on during ZOOM or other meetings which reduced their sense of participation? Some further discussion of technology and how this can democratise or exclude in particular would be appreciated.

9. Table 2 is a good summary – however I wonder if there is a role for any ‘rest stops’ in the study process where team members take time for discussion about the coproduction process – any points to elicity explicit reflexivity about positions/ power relations and how these influence interactions within the team as well as CBR with study participants. See the Schaaf paper you have cited and perhaps hold up your process to consider whether it has fully engaged with all the ways power relations can influence coproduction and participation.

10. Discussion – can you please elaborate on how CBRs participated in analysis/ paper writing given English medium and academic content – e.g. thematic analysis/ discussion

6. PLOS authors have the option to publish the peer review history of their article (what does this mean?). If published, this will include your full peer review and any attached files.

**Do you want your identity to be public for this peer review?** For information about this choice, including consent withdrawal, please see our Privacy Policy.

Reviewer #1: **Yes: **Anna Madill a.l.madill@leeds.ac.uk

Reviewer #2: **Yes: **Dr Kaaren Mathias

---

## [Decision Letter · Decision Letter 1]

21 Sep 2022

Cultivating capacities in community-based researchers in low-resource settings: Lessons from a participatory study on violence and mental health in Sri Lanka

PGPH-D-22-00472R1

Dear Dr Palfreyman,

We are pleased to inform you that your manuscript 'Cultivating capacities in community-based researchers in low-resource settings: Lessons from a participatory study on violence and mental health in Sri Lanka' has been provisionally accepted for publication in PLOS Global Public Health.

Best regards,

Prashanth Nuggehalli Srinivas, MBBS, MPH, PhD

Academic Editor

Reviewer Comments (if any, and for reference):

Reviewer's Responses to Questions

**Comments to the Author**

1. If the authors have adequately addressed your comments raised in a previous round of review and you feel that this manuscript is now acceptable for publication, you may indicate that here to bypass the “Comments to the Author” section, enter your conflict of interest statement in the “Confidential to Editor” section, and submit your "Accept" recommendation.

Reviewer #1: All comments have been addressed

Reviewer #2: All comments have been addressed

2. Does this manuscript meet PLOS Global Public Health’s publication criteria? Is the manuscript technically sound, and do the data support the conclusions? The manuscript must describe methodologically and ethically rigorous research with conclusions that are appropriately drawn based on the data presented.

Reviewer #1: Yes

Reviewer #2: Yes

3. Has the statistical analysis been performed appropriately and rigorously?

Reviewer #1: N/A

Reviewer #2: N/A

4. Have the authors made all data underlying the findings in their manuscript fully available (please refer to the Data Availability Statement at the start of the manuscript PDF file)?

Reviewer #1: No

Reviewer #2: No

5. Is the manuscript presented in an intelligible fashion and written in standard English?

Reviewer #1: Yes

Reviewer #2: Yes

6. Review Comments to the Author

Reviewer #1: Thank you for addressing my comments. I am happy that this has been done thoroughly. In my opinion, the article - which addresses also reviewer 2's comments - is clear and a very helpful example from which others can also learn.

Reviewer #2: (No Response)

7. PLOS authors have the option to publish the peer review history of their article (what does this mean?). If published, this will include your full peer review and any attached files.

**Do you want your identity to be public for this peer review?** For information about this choice, including consent withdrawal, please see our Privacy Policy.

Reviewer #1: No

Reviewer #2: **Yes: **Kaaren Mathias
